# The Effect of High-Energy Ionizing Radiation on the Mechanical Properties of a Melamine Resin, Phenol-Formaldehyde Resin, and Nitrile Rubber Blend

**DOI:** 10.3390/ma11122405

**Published:** 2018-11-28

**Authors:** Ivan Kopal, Juliana Vršková, Ivan Labaj, Darina Ondrušová, Peter Hybler, Marta Harničárová, Jan Valíček, Milena Kušnerová

**Affiliations:** 1Faculty of Industrial Technologies in Púchov, Alexander Dubček University of Trenčín, Ivana Krasku 491/30, 020 01 Puchov, Slovakia; juliana.vrskova@fpt.tnuni.sk (J.V.); ivan.labaj@fpt.tnuni.sk (I.L.); darina.ondrusova@fpt.tnuni.sk (D.O.); 2Progresa Final SK, s.r.o., Ferienčíková 18, 811 08 Bratislava, Slovakia; peter.hybler@progresafinal.sk; 3Faculty of Engineering, Slovak University of Agriculture in Nitra, Tr. A. Hlinku 2, 949 76 Nitra, Slovakia; marta.harnicarova@uniag.sk or harnicarova@mail.vstecb.cz (M.H.); jan.valicek@uniag.sk or valicek.jan@mail.vstecb.cz (J.V.); 4Institute of Technology and Business in České Budějovice, Faculty of Technology, Department of Mechanical Engineering, Okružní 10, 370 01 České Budějovice, Czech Republic; kusnerova.milena@mail.vstecb.cz

**Keywords:** high-energy ionizing radiation, polymer modification, mechanical properties, polymer blends, polymer friction composite systems

## Abstract

Irradiation by ionizing radiation is a specific type of controllable modification of the physical and chemical properties of a wide range of polymers, which is, in comparison to traditional chemical methods, rapid, non-polluting, simple, and relatively cheap. In the presented paper, the influence of high-energy ionizing radiation on the basic mechanical properties of the melamine resin, phenol-formaldehyde resin, and nitrile rubber blend has been studied for the first time. The mechanical properties of irradiated samples were compared to those of non-irradiated materials. It was found that radiation doses up to 150 kGy improved the mechanical properties of the tested materials in terms of a significant increase in stress at break, tensile strength, and tensile modulus at 40% strain, while decreasing the value of strain at break. At radiation doses above 150 kGy, the irradiated polymer blend is already degrading, and its tensile characteristics significantly deteriorate. An radiation dose of 150 kGy thus appears to be optimal from the viewpoint of achieving significant improvement, and the radiation treatment of the given polymeric blend by a beam of accelerated electrons is a very promising alternative to the traditional chemical mode of treatment which impacts the environment.

## 1. Introduction

The modification of construction materials, including polymers, is inevitable, particularly in order to achieve properties meeting all the requirements of increasingly-demanding practical applications [1]. At present, a way to tailor the properties of polymeric materials in industrial practice is their treatment with ionizing beta radiation generated in electron accelerators, which are capable of producing high radiation doses in a relatively very short time. High-energy ionizing radiation can modify the macroscopic properties and molecular structure of the irradiated polymers [2]. During electron beam irradiation (EB), the energy of accelerated electrons is injected directly into a material, in which it subsequently induces chemical reactions, often without the need for any catalyst. The processes initiated by EB radiation in the polymer are very complex, as they are accompanied by several simultaneously running, competitive reactions, such as cross-linking, degradation, oxidation, fragmentation, grafting, and others [3]. However, the most useful reaction occurring during polymer irradiation is a radiation-induced cross-linking of polymer chains, which is capable of improving some physical and chemical properties, such as hardness, resilience, thermal resistance, solubility, and several others [4]. For a relatively large group of polymers modified by high energy EB radiation, the improvement of these properties is even more substantially pronounced than that achievable by conventional cross-linking [5]. In general, EB radiation causes the excitation of polymeric macromolecules occurring adjacent to the incident electrons. The energies associated with the excitation of these molecules are proportional to the electron velocities and the radiation dose [6]. The interaction of incident electrons with molecules leads to ionization in the polymer and numerous highly-reactive forms are produced (such as free neutral radicals, radical cations and anions, low energy electrons, and singlet and triplet states of molecules excited by electrons) in a whole series of processes (such as fragmentation to carbocations and free radicals, capture of electrons by polymeric and oxygen molecules, dissociative capture of electrons, and others) [7]. Free radicals formed by dissociation of the molecules in their excited state or by the interaction of molecular ions, as well as molecular ions, can react by linking the polymer chains directly to the 3D network structure or by initiating grafting reactions. Due to the full depth of penetration of the incident electrons, EB radiation results in uniformly-cured polymers [8]. Chain branching and cross-linking increase the molecular weight of the polymer, while degradation or scission causes a reduction of the initial molecular weight. During irradiation, both these phenomena coexist, and their prevalence depends on several factors, such as the initial molecular structure and morphology of the polymer, and the radiation conditions. The EB radiation-induced cross-linking will result in an increase in physical and chemical properties until chain scission and breaking of intermolecular bonds reduces them; therefore, finding an optimal radiation dose is always necessary in order to prevent polymer degradation caused by excessive irradiation [8,9].

At present, a number of research papers are dedicated to the modification of polymers by EB radiation. The fundamental principles of all applications of radiation treatment of polymers are evaluated, for example, in [10]. In general, it is well known that some polymers can be cross-linked with EB radiation, while others tend to degrade. Some of them are capable of self-cross-linking, while some have to be first mixed with a cross-linking agent, and EB modification is applied only during their polymerization process, which is presented in a number of cases in [11]. Ref. [12] gives a brief overview of polymers modifiable and non-modifiable by ionizing radiation. The aim of the paper [13] is to balance the applications of radiation cross-linking of polymers in chemically-aggressive and high-temperature conditions. Ref. [14] studies the mechanical and thermal properties of the epoxy resin irradiated by EB irradiation with doses up to 300 kGy; its authors deal with the process of curing with EB irradiation, using cationic initiators of the cross-linking reaction. The result of radiation curing is compared with the result of the process of heat curing that is used in a standard manner for this type of polymers. The samples of the investigated material were irradiated by EB irradiation for several minutes with a total dose of 150 kGy. The heat curing process was carried out for 20 h in total. The glass transition temperatures, *T_g_*, determined by the DMA (dynamic mechanical analysis) technology, achieved values higher even by 17 per cent in irradiated samples compared to heat-cured samples, due to the higher density of the formed polymeric 3D network resulting in higher stiffness of the material cured by irradiation. The authors of this paper concluded that curing of epoxy resin by EB irradiation may be particularly suitable for applications in which high thermal stability and heat resistance are required. Ref. [15] deals with the study of the thermal and mechanical properties of epoxy-diane resin cured with polyethylene-polyamine after irradiation by doses of 30, 100, and 300 kGy. The effects of individual radiation doses were investigated using standard static tensile tests, as well as with the use of TGA (thermogravimetric analysis) and DMA methods. It was found that the thermal properties of the investigated material slightly increased after EB irradiation. Mechanical properties increased after irradiation by doses of 30 kGy and 100 kGy, while at a dose of 300 kGy, they significantly decreased. The impact of EB irradiation on the mechanical and dynamic mechanical properties of cross-linked fluorocarbon rubber, natural rubber, ethylene-propylene-diene monomer of rubber, and nitrile rubber was investigated [16]. It was found that the modulus of elasticity, gel portion, the temperature of glass transition, and dynamic elastic modulus of the investigated vulcanizates increase with the increasing irradiation dose, while the elongation at break and the loss factor decrease. The authors of the paper [17] studied the mechanism of interaction between carbon fibers and phenol epoxy matrixes of polymeric composites cured by EB radiation, as well as improving their interfacial shear strength. Promising results were obtained by electrochemical processing of carbon fibers followed by subsequent application of a reagent compatible with the applied EB. The morphology of both modified and unmodified carbon fibers was characterized by the use of SEM (scanning electron microscopy), AFM (atomic force microscopy), and XPS (X-ray photoelectron spectroscopy). It was proved that acidic electrolytes are particularly well-suited to improving interfacial adhesion in the EB curing process. Alkaline groups on carbon fibers prevent cationic polymerization and improve the shear strength of EB-cured composites.

The majority of relevant papers publish the results of research on the impact of high-energy EB irradiation on the micromechanical, thermo-mechanical, visco-elastic, and rheological properties of various types of polymeric materials, their mixtures and composites, nanocomposites based on polymeric matrixes, as well as the influence of irradiation on changes in their structure, degree of cross-linking, crystallinity, and gel content under various conditions of radiation exposure, typically up to 300 kGy, using the most advanced analytical tools for modern, standard mechanical tests: TGA, DMA, and DSC (differential scanning calorimetry) techniques, FT-IR (Fourier-transform infrared spectroscopy) and Raman spectroscopy, SEM, AFM, or XPS. However, to the best of our knowledge, no previous studies have been reported describing the effect of EB irradiation on the polymer blend of melamine resin, phenol-formaldehyde resin, and nitrile rubber, representing a specific three-component mixture of two reactoplastics and one elastomer, as well as its individual components. The models of the results of an interaction of ionizing radiation with polymeric systems are virtually completely missing. For this reason, the aim of the present paper is to investigate and model the influence of EB irradiation on the mechanical properties of the aforementioned polymeric composition which are commonly used as a polymeric matrix of friction composite systems in a number of practical applications, particularly in the automotive industry [18].

## 2. Materials and Methods

### 2.1. Preparation of Samples

The PMX3 polymer system (PS) was purchased from a professional material manufacturer for polymer matrices of friction composite systems [19]. PS PMX3 consists of a mixture of melamine resin, phenol-formaldehyde resin, and nitrile rubber, which is distributed by the manufacturer (Ahshenhui, Jinhu, China) in the form of granules as a polymer blend with a proportion of ingredients adapted for wider use in the automotive industry. PS PMX3 was plasticized for 15 min at 110 ± 2 °C and pressure (80 ± 1) kN in a Fontijne Presses vulcanization press, LabEcon Series 600 (Fontijne, Barendrecht, The Netherlands), followed by homogenization using a laboratory double roller VOGT LaboWalz W 80 T (VOGT, Berlin, Germany) into plates of thickness (2 ± 0.05) mm. Prior to further processing, PS PMX3 was allowed to stand for 24 h at 25 ± 2 °C. Pieces in the shape of a two-sided blade in accordance with standard ISO 37 [20] or ASTM D 412 [21], were cut out from the prepared plates by a pneumatic cutter CEAST, Hollow Punch-pneumatic 6054.000 (Instron, Bucking-Hamshire, UK). The dimensions of the test pieces (Type 2) were as follows: overall length (minimum)—75 mm, width of ends—(25 ± 1) mm, length of narrow portion—(25 ± 1) mm, width of narrow portion—(4 ± 0.1) mm, transition radius outside—(8 ± 0.5) mm, transition radius inside—(12.5 ± 1) mm.

### 2.2. Radiation Treatment

Irradiation of the samples was carried out using a linear electron accelerator UELR-5-1S (NIIEFA, Sankt-Peterburg, Russia) with a vertical design of the construction. The source of electrons in the accelerator is an indirectly-heated, pressed Ba-Ni cathode with a diameter of 5 mm. The emitted electrons are formed by the electron-optical system designed by J.R. Pierce. The increase in the kinetic energy of the electrons is ensured by the electric field of the stationary electromagnetic wave produced by a magnetron operating in the pulse mode at a frequency of 2998 MHz. The focusing of the beam is realized by the high-frequency field. The output device (scanning chamber) provides the ebeam with a band of lengths 400 mm, 450 mm and 500 mm. The beam outlet to the atmosphere is through 50 μm thick titanium foil. The dispatch of the beam is realized in a vacuum chamber by a scanning electromagnet.

The technical parameters of the accelerator during the experiments were the following: the energy of electrons (5 ± 5%) MeV, pulse duration (3.5 ± 0.5) μs, beam diameter at the outlet of the foil ≥2 mm. Homogeneous irradiation of the samples of the investigated PS PMX3 was ensured in a dynamic manner, i.e., that the irradiated samples were uniformly moving on the conveyor under the beam at a defined velocity. The experiments were carried out in the air, at normal pressure, and at room temperature. Excessive overheating due to higher doses of irradiation was prevented by the placement of the samples on heat-insulating materials and by air flowing from the ventilation system. The ventilation system also provided an extraction of the ozone generated during irradiation.

The samples were irradiated with a wide span of target doses, namely: 77, 138, 150, 180, 190, and 284 kGy. The doses above 150 kGy, which were above the maximum limit of dosimetric systems, were executed by multiple irradiations. The break between two exposures did not exceed 20 min. The total irradiation time of the samples at individual radiation doses did not exceed 54 min. Due to the required high radiation doses, the highest accelerator outputs corresponding to pulse frequencies of ebeam the 120 Hz and 240 Hz bands were used. The dose depends on the speed of the conveyor section below the accelerator window, and this dependence is non-linear and is determined experimentally. Calculation of radiation doses was performed by a routine dosimetric system using radiochromic B3 foils (GEX Corporation, Centennial, CO, USA). The foils cut to circles with a diameter of 1 cm react to radiation by showing a change of color. The absorption coefficients of the irradiated foils were measured by spectrophotometer (Genesys20, Thermo Electron Corporation, Madison, WI, USA). The radiation dose is calculated from the experimentally-determined dependencies of the dose on absorbance. The combined uncertainty of the applied dosimetric system is ≤6%.

### 2.3. Static Uniaxial Tensile Tests

The tensile properties of the samples of the investigated material were determined using a computer-controlled universal tensile tester Shimadzu AG-X Plus shifter (Shimadzu, Tokyo, Japan) with two pneumatic clamping jaws with a mutual movement speed of (100 ± 5) mm·min^−1^ at room temperature. The mean value of at least five dumbbell specimens of each sample, prepared in accordance with standards ISO 37 or ASTM D 412 [20,21], was taken, although specimens that broke in an unusual manner were disregarded. The average engineering stress-strain curves were constructed from the obtained experimental average force vs. elongation data with a maximum relative standard deviation of 5%, and they were analyzed using the software package Matlab^®^ Version 7.10.0.499 R2010a 64-bit (MathWorks, Natick, MA, USA).

## 3. Results and Discussion

### 3.1. Changes in Mechanical Properties

The results of the uniaxial tensile tests for samples of the non-irradiated PS PMX3 and PS PMX3 irradiated with high energy EB radiation by doses ranging from 77 kGy to 284 kGy, in the form of average engineering stress-strain curves, are presented in Figure 1a. As expected, the effect of EB irradiation on the shape of the stress-strain curves, obtained under the given conditions of tensile tests, is evident. It is apparent already at a first glance that all curves exhibit a relatively short linear elastic region with a low proportionality ratio, a relatively short linear visco-elastic region with low value of limit of elasticity, a relatively low value of tensile modulus and ultimate strength, as well as tear stress, but that their strain at break achieves an extremely wide range of values. A detailed view of the linear region of the stress-strain curves is shown in Figure 1b.

The shape of the stress-strain curve of the virgin sample resembles the stress-strain curve of the linear amorphous reactoplastics (non-cross-linked resins) with low strength, without an ultimate strength, but with ductility of non-cross-linked, non-crystallizing elastomers in a rubbery state near viscous flow temperature [22]. Once the strength limit has been reached, the stress decreases with the increasing strain in a non-linear manner, albeit continuously. The material ductility at the same time was so big that its sample during the performed tensile tests could not be ruptured even at a strain above 500%. The stress-strain curves after irradiation of material with radiation doses of 77 kGy and 138 kGy show the characteristics of partially cross-linked non-crystallising elastomers in a rubbery state high above the temperature of their glass transition *T_g_* [23], with much higher strength and lower ductility than in the non-irradiated sample. 

As a result of irradiation of the sample with a radiation dose of 150 kGy, its stress-strain curve showed a shape corresponding to the tensile response of a brittle, three-dimensional (already cured) amorphous resin in a glassy state, well below the *T_g_* temperature [24], with ductility smaller than in the case of application of lower radiation doses. The strength of the irradiated material at a dose of 150 kGy achieves the maximum value at the same time. The shape of the stress-strain curve at higher radiation doses corresponds to curves of the amorphous cured resins in a glassy state, at a temperature just below *T_g_* [22], with strength substantially lower than at a dose of 150 kGy, and with a corresponding ductility.

It is evident from the above that the tensile response of the virgin sample reflects the mechanical properties of the polymeric blend of non-cross-linked resins and rubber forming PS PMX3. After irradiation with radiation doses below 150 kGy, the mechanical properties of the partially cross-linked rubber predominate. At a dose of 150 kGy, the properties of the cured resins are dominant, whereas, at higher radiation doses, the effects of radiation-induced degradation of a polymeric network of irradiated material are manifested with predominant mechanical properties of the cured resins. During EB irradiation with radiation doses within the given range of values, the irradiated material progressively enters various physical states undergoing various relaxation transitions events initiated by the absorbed ionizing radiation, resulting in observed non-linear changes in its mechanical properties. A more detailed analysis of the stress-strain curves makes it possible to quantify the effect of the magnitude of radiation dose on the basic mechanical characteristics of the investigated PS PMX3, such as stress and strain at break, ultimate limit, and the tensile modulus *M* 40. The tensile modulus at 40% strain was selected for assessing the stiffness of the irradiated material in the initial stages of deformation mainly because of the possibility of its identification for all applied radiation doses to the break of the sample and a good information capability on the elasticity of tested material [23].

The values of strain at break *ε*_b_ and of the tensile stress at break *σ*_b_ of the investigated PS PMX3 for different radiation doses, in the form of stress-strain curves to the break of the sample, are presented in Figure 2a, while their changes with the increasing dose of irradiation are shown in Figure 2b. Vertical abscissas show the error intervals of experimental data with the corresponding standard deviation. It is evident from Figure 2a,b that up to a dose of 150 kGy, *ε*_b_ decreases in a non-linear manner, but continuously, with a corresponding increase of *σ*_b_. Changes in values *ε*_b_ and *σ*_b_ after irradiation of the samples of PS PMX3 by radiation doses of 77 kGy, 138 kGy, and 150 kGy thus show the expected opposite trend. The value *ε*_b_ = 180.9% at 150 kGy versus the value *ε*_b_ = 332.4% at a dose of 138 kGy is approximately 1.84 times lower, and compared to the value *ε*_b_ = 426% at 77 kGy, it is approximately 2.36 times lower. The value *σ*_b_ = 7.681 MPa at 150 kGy is versus the value *σ*_b_ = 4.441 MPa at 138 kGy more than 1.73 times higher, and compared to the value *σ*_b_ = 2.466 MPa at 77 kGy, it is approximately 3.12 times higher. The identified changes of values *ε*_b_ and *σ*_b_ demonstrate the changes in the mechanical properties of the irradiated PS PMX3 in this interval of radiation doses in terms of a significant reduction in its total ductility (*ε*_b_), while at the same time, its tensile strength (*σ*_b_) significantly increases due to the prevalence of cross-linking reactions over the intensity of the reactions of polymer chain scission and degradation of intermolecular cross-links initiated by the absorbed high-energy EB radiation [25]. Higher radiation doses produce more cross-links between macromolecular chains of irradiated material, resulting in an increase in resistance to the release of intermolecular forces by mechanical loading, or in an increase in its stiffness. At higher radiation doses, the polymeric network becomes tighter, and it restricts the internal mobility of the chains, thereby increasing the resistance of the irradiated material to its mechanical damage, or its strength [26]. Higher density of 3D structures of polymeric network and limited mobility of polymeric chains, i.e., higher stiffness, as well as strength of material due to irradiation by higher radiation doses, also prevent the structural reorganization of polymeric chains during its tensile stress, which significantly reduces the ability of PS PMX3 to be plastically deformed by drawing, or its ductility [27]. At radiation doses above 150 kGy, the dependence of both *ε*_b_ and *σ*_b_ on the magnitude of the dose of the absorbed EB radiation is non-monotonic. At 180 kGy *σ*_b,_ it drops to the value of 6.637 MPa, while *ε*_b_ rises to 281.2%; at 190 kGy, *σ*_b_ and *ε*_b_ simultaneously drop to 3.172 MPa and 54.39%, while at a dose of 284 kGy, *σ*_b_ and *ε*_b_ simultaneously increase to the values of 5.041 MPa and 107.5%, respectively. At a dose of 180 kGy, due to the dominance of the degradation processes over the radiation, cross-linking prevails over the degradation of transverse inter-molecular bonds between the chains of the polymeric network over their production, which makes the network less tight. The lower density of bonds reduces the size of intermolecular interactions and releases limited mobility of chains, which is accompanied by observed decreases in strength (*σ*_b_) and increased ductility (*ε*_b_) of the irradiated material. During further increases of the radiation dose, the changes of *ε*_b_ and *σ*_b_ show the same trend with an analogous course, so it is possible to assume that further degradation of the radiation-induced polymeric network of the irradiated material will occur. The *ε*_b_ and *σ*_b_ values for the non-irradiated sample could not be determined, as it was not ruptured under the given conditions of the tensile tests.

The values of the strength limit *σ*_m_ and the modulus *M* 40 of the investigated PS PMX3 for the virgin sample and the samples irradiated with different doses of EB radiation are presented in Figure 3a, and their changes with the increasing dose of absorbed radiation are shown in Figure 3b. For clarity, Figure 3a shows the entire stress-strain curves till completion of the tensile tests, not only after they have been torn apart, as in Figure 2a. It is evident from Figure 3a,b that up to a dose of 150 kGy both *σ*_m_ and *M* 40 are non-linear, but they grow monotonously. The values *σ*_m_ = 3.23 MPa at 77 kGy, *σ*_m_ = 4.734 MPa at 138 kGy and *σ*_m_ = 7.674 MPa at 150 kGy, compared to the non-irradiated sample *σ*_m_ = 2.143 MPa, are approximately 1.5, 2.1, and 3.6 times higher, respectively. Values of the modulus *M* 40 = 2.598 MPa at 77 kGy, *M* 40 = 3.059 MPa at 138 kGy and *M* 40 = 4.3110 MPa at 150 kGy, compared to *M* 40 = 2.13 MPa for a non-irradiated sample, are approximately 1.2, 1.4, and 2 times higher, respectively. The recorded changes in the values of *σ*_m_ and *M* 40 after application of radiation doses of 77 kGy, 138 kGy, and 150 kGy demonstrate the expected increase in tensile strength of the irradiated PS PMX3 with a simultaneous increase in stiffness at the initial stages of deformation, which can also be attributed to the prevailing radiation-induced cross-linking reactions over the degradation processes in the irradiated material, with the mechanism described above. At radiation doses above 150 kGy, as with at *ε*_b_ and *σ*_b_, the dependencies of both *σ*_m_ and *M* 40 on the magnitude of the dose of the absorbed radiation are non-monotonic, but they exhibit the same trend with an analogous course. At 180 kGy, the value *σ*_m_ drops to 6.682 MPa and *M* 40 drops to 3.67 MPa; at 190 kGy, the value *σ*_m_ drops to 3.476 and *M* 40 drops to 3.21 MPa, due to the induction-induced reduction of strength and an increase of the ductility of the sample due to the predominance of the degradation processes over the forming processes of the polymer network with the mechanism described above. At a dose of 284 kGy, the values of *σ*_m_ and *M* 40 simultaneously increase to 5.551 MPa and 3.931 MPa, respectively. However, this increase in *σ*_m_ and *M* 40, as well as the increase of *ε*_b_ and *σ*_b_ at the radiation dose of 284 kGy, is due to the continued degradation of the sample material, and not to the subsequent process of formation of the polymeric network or by the radiation-induced change of crystallinity of PS PMX3 [28]. Due to the chemical composition of the irradiated material, it can be assumed that at high radiation doses, the cross-linked nitrile rubber therein becomes completely disintegrated, and it thereby transforms to a filler dispersed in the polymeric matrix of the blend of melamine and phenol-formaldehyde resin of the composite created by the high-energy EB irradiation with mechanical properties quite different from those of the original polymer system. However, this assumption will need to be verified in ongoing research using the DSC, DMTA, TGA, and FT-IR techniques, or other diagnostic techniques.

The ultimate strength after irradiation of the investigated material with a radiation dose of 150 kGy is equal to the stress at break of *σ*_b_, which is typical for brittle amorphous polymers, such as cross-linked reactoplasts, including the cured resins [24]. During all other applied radiation doses, the tensile response of the irradiated PS exhibits signs of the tensile response of more ductile materials with significantly lower stiffness and strength [23]. From the point of view of enhancement of mechanical properties of the investigated PS PMX3 with its irradiation by high-energy EB radiation, it is possible to consider the radiation dose of 150 kGy as optimal.

The experimental values of the monitored mechanical characteristics for the virgin sample, as well as for the PS PMX3 samples irradiated with the individual radiation doses, are summarized in Table 1.

### 3.2. Regression Analysis

The effect of EB irradiation on the change of mechanical properties of PS PMX3 by radiation doses up to 150 kGy, at which it is possible to observe a significant improvement, is demonstrated by results of the regression analysis at the confidence level of 95% shown in Figure 4a,b (for *ε*_b_ and *σ*_b_), and in Figure 5a,b (for *σ*_m_ and *M* 40).

The realized regression analysis of the experimental data has shown that all the analyzed functional dependencies at a given interval of radiation dose can be described with a relatively high degree of reliability by a single regression model, in the form
(1)$$y\left(x\right)\;=\;\sum_{i\;=\;1}^{N}\Delta y_{i}\;\exp\;\left(\mp \;{\left(\frac{x}{\theta_{i}}\right)}^{m_{i}}\right)\;+\;\delta \;,$$
where *y*(*x*) is the corresponding mechanical characteristic of the irradiated material, *x* is the radiation dose absorbed, and Δ*y_i_*, *θ_i_*, *m_i_,* and *δ* represent unknown but the correct coefficients of the model that have been reliably estimated in the process of parametric fitting of experimental data using the Trust Region algorithm of non-linear least squares method [29] in the device ‘CF Tool’ of the Matlab^®^ software package. The following is then valid for the model coefficients Δ*y_i_*:(2)$$\sum_{i\;=\;1}^{N}\Delta y_{i}\;=\;y\left(0\right)\;-\;\delta ,\;$$
where *y*(0) is the initial value of the corresponding mechanical characteristic as it follows from the relation (1) for *x* = 0. The subscript *_i_* pertains to critical points of the curve *y = y* (*x*) where a significant change in the velocity of its trend takes place, and *N* is the number of these critical points. The negative value of the quotient in the model exponent then represents a decreasing trend, while its positive value represents the increasing trend of *y*(*x*). The coefficient *δ* represents a non-constant error parameter of the model, which will be discussed later.

The dashed lines in Figure 4 and Figure 5 represent the 95% prediction intervals of the estimated coefficients of the model representing the 5% probability that the results of future measurements of the mechanical characteristics of the irradiated PS PMX3 under the same experimental conditions will not lie between their lower and upper boundaries [30].

The regression model (1) describes changes in the observed mechanical characteristics of the irradiated polymeric material due to the prevalence of formation of transverse intermolecular bonds between macromolecular chains and their simultaneous breaking accompanied by degradation of the formed polymer network. Due to the different nature of intermolecular bonds present in the polymer (hydrogen, dipole, Van der Waals, or ionic interactions with different values of dissociation energy), as well as due to the different spatial arrangement of macromolecules, a specific distribution of intermolecular bond strengths is observed between macromolecules of polymeric materials, which match the Boltzmann distribution. However, radiation-induced (similarly as temperature-induced) breaking of intermolecular bonds is not a random process subject to Boltzmann’s distribution [24]. Since the failure of each bond leads to a redistribution of stresses in all other bonds, each previous failure event has a notable effect on the failure of all remaining bonds. The interaction of these failures is therefore subject to Weibull’s statistics [31]. Given the above, the regression model (1) can be interpreted in a good approximation by the function of probability density of the Weibull’s distribution of EB radiation-induced breaking of transverse intermolecular bonds of the irradiated polymer in the presence of simultaneous formation of a polymeric 3D network with new cross-links (quantified by an error *δ* of the model) in the following form [32]:(3)$$y\left(x\right)\;=\;bkx^{k-1}e^{-\;bx^{k}},$$
where the pre-exponential factor or hazard function *bkx^k^*^−1^ of the Weibull’s distribution approximates the constants Δ*y_i_*, the scale parameter *b* = ±1/*θ_i_* and the shape parameter, or the Weibull modulus *k* = *m_i_*. Since the sum (ΣΔ*y_i_* + *δ*) approximates the initial values of the respective mechanical characteristics *y*(0) and *θ_i_* of the radiation dose at the critical points of the curve *y* = *y*(*x*), the values Δ*y_i_* and *θ_i_* depend on the initial physical state of the irradiated sample of material, its properties and conditions of irradiation, while the Weibull’s moduli *m_i_* reflect the statistics of transverse intermolecular bonds breaking due to EB irradiation with different radiation doses. Since the breaking of bonds is conditioned by overcoming the energy barrier of intermolecular forces, the values of the coefficients *m_i_* depend on the magnitude of the activation energy of the individual relaxation events observed at the applied radiation doses due to the release of the polymeric chains with respect to the relevant type of motion corresponding to the given primary as well as secondary relaxation event [33]. At the same time, the degree of chains mobility is determined by the actual physical state of the polymer. In the glassy state, it is limited only to local movements of individual molecules, vibrations, bending and stretching of bonds, rotation of lateral molecular groups, and movement of only a few main chains. In the area of the glass transition, significant movement of lateral groups is possible, as well as gradual movement and reptation of the main chains, which transforms to a large-scale movement of segments and whole chains in a rubbery state. The state of viscous flow allows for the sliding of whole macromolecular chains and global translations of entire polymer molecules between entanglements with subsequent decay of the polymeric material as a whole [34]. Since the critical condition for radiation-induced polymer cross-linking is, in addition to the formation of secondary radicals in its amorphous regions, also the sufficiently high mobility of the chains which are carried by these secondary radicals, the radiation cross-linking by ionizing radiation is an optimal rubbery, or visco-elastic state with high mobility of entire chains [28]. Since the sensitivity of the individual tensile characteristics *ε*_b_, *σ*_b_, *σ*_m_, and *M* 40 to the magnitude of the absorbed radiation dose is different, it can be naturally expected that values of the parameters *m_i_* for their description with the use of the model (1) will also be different.

The radiation cross-linking is an excessively-complicated process due to the complexity of the polymerization reactions running in the irradiated three-component material, the analytical description of which requires a solution of the system of non-linear differential equations. This solution can be obtained only with the use of numerical methods with an approximate result [34]. In order to simplify this problem, the influence of the radiation-induced formation of the polymeric network on the mechanical properties of the irradiated PS PMX3 in the model (1) is approximated only by an unknown, non-constant error parameter *δ*. The value of the error parameter at the same time depends on the magnitude of the radiation dose *x*, as well as on the values of other coefficients of the model Δ*y_i_*, *θ_i_*, and *m_i_*, i.e., on the initial physical state of the irradiated material sample, its properties and conditions of irradiation, as well as on the activation energies of the relaxation events initiated by individual radiation doses of the absorbed radiation that together determine the intensity of radiation-induced generation of cross-links and formation of polymer network.

The coefficients Δ*y_i_*, *m_i_* and *δ*, estimated in the process of parametric fitting of the experimental data, coefficients *θ_i_* and percentage deviations of the coefficients Δ*y*(0) from their experimental values, as well as the statistical parameters ‘goodness of fit’ SSE, RMSE, R^2^ and Adj-R^2^ [35], are listed in Table 2 (for *ε*_b_ and *σ*_b_) and Table 3 (for *σ*_m_ and *M* 40). Due to the aforementioned low number of available experimental data, it was impossible to identify the coefficients *θ_i_* by parametric fitting of the experimental data; therefore, they were estimated directly from the graphical interpretation of the functional dependencies of the monitored mechanical characteristics of the irradiated material.

Relatively low values of SSE and RMSE, the close proximity of the parameters R^2^ and Adj-R^2^ to one (or their equality at the value of one), and low values of deviations Δ*y*(0) show the relatively high performance of the found model. However, due to the low amount of available experimental data, more detailed statistical analysis of the descriptive and predictive capabilities of the model was not possible, and it will be necessary to realize this within the framework of ongoing research.

## 4. Conclusions

In this paper, for the first time, the effect of high-energy ionizing radiation on the mechanical properties of a melamine resin, phenol-formaldehyde resin, and nitrile rubber blend, commonly used as a polymeric matrix of friction composite systems, especially in the automotive industry, was studied. The study addressed the changes in the tensile characteristics of the respective blend after irradiation with a 5 MeV electron beam, using the radiation doses ranging from 77 kGy to 284 kGy. The results of standard tensile tests have shown that irradiation with radiation doses up to 150 kGy induced formation of a spatial polymeric network, and as a consequence, the values of stress at break, ultimate strength, and tensile modulus at strain of 40% significantly increase with the increasing dose of radiation until the value of strain at break of the irradiated blend decreases. At higher radiation doses, the polymeric network formed as a result of irradiation already starts to degrade, and the monitored tensile characteristics of the blend are deteriorating. The radiation dose of 150 kGy thus appears to be optimal from the viewpoint of their significant improvement and the radiation treatment of the given polymeric blend by the beam of accelerated electrons as a potential alternative to the traditional chemical mode of treatment which burdens the environment.

The functional dependence of the monitored tensile characteristics on magnitude of the radiation dose with a value of up to 150 kGy was analyzed with a high degree of reliability by means of a new, physically-based model based upon the statistics of Weibull’s distribution of transverse intermolecular bonds of polymeric chains breaking under the influence of ionizing radiation with an error parameter quantifying the radiation-induced simultaneous formation of the polymeric network.

## Figures and Tables

**Figure 1 materials-11-02405-f001:**
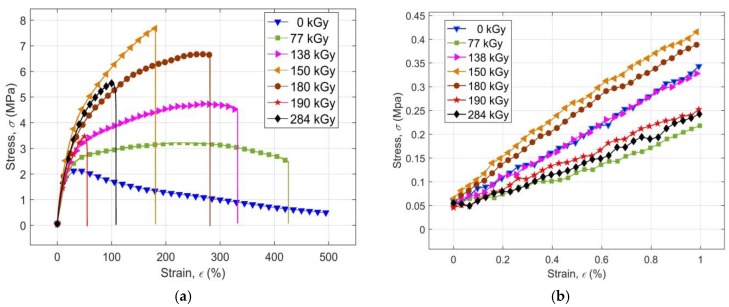
(**a**) Experimental average engineering stress-strain curves of non-irradiated polymeric system PMX3 and of polymeric system PMX3 modified by various radiation doses of ionizing EB radiation; (**b**) Detailed representation of the linear region of the stress-strain curves.

**Figure 2 materials-11-02405-f002:**
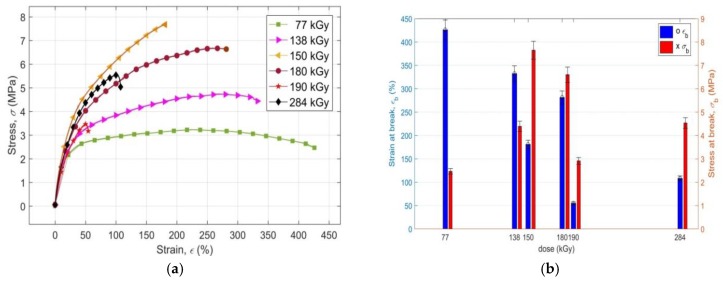
(**a**) Presentation of the values of strain at the break *ε_b_* and of stress at break *σ_b_* of the polymeric system PMX4 for various radiation doses; (**b**) The effect of the radiation dose on the values of strain at break *ε_b_* and stress at break *σ_b_* of the polymeric system PMX3.

**Figure 3 materials-11-02405-f003:**
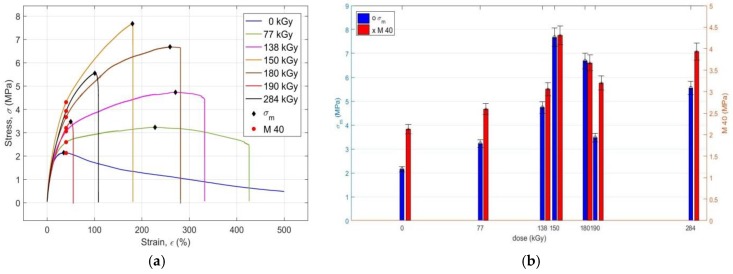
(**a**) Presentation of the strength limit values *σ_m_* and the modulus *M* 40 of the polymeric system PMX3 for the virgin sample and for the samples irradiated with different radiation doses; (**b**) Effect of the radiation dose on the strength limit values *σ_m_* and the modulus *M* 40 of the polymeric system PMX3.

**Figure 4 materials-11-02405-f004:**
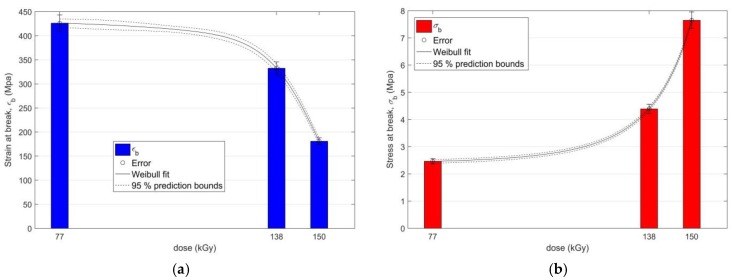
(**a**) Regression analysis of the functional dependence of strain at break *ε_b_* on the radiation dose; (**b**) Regression analysis of the functional dependence of stress at break *σ_b_* on the radiation dose.

**Figure 5 materials-11-02405-f005:**
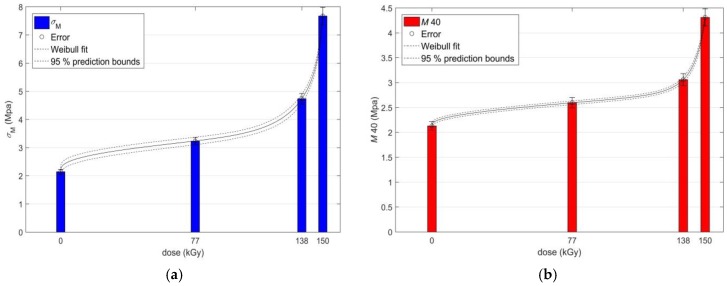
(**a**) Regression analysis of the functional dependence of the strength limit *σ_m_* on the radiation dose; (**b**) Regression analysis of the functional dependence of the modulus *M* 40 on the radiation dose.

**Table 1 materials-11-02405-t001:** Experimental values of mechanical characteristics *ε*_b_*, σ*_b_, *σ*_m_ and *M* 40 for the virgin sample of PMX3 polymer system and samples irradiated with individual radiation doses.

Mechanical Characteristics	0 kGy	77 kGy	138 kGy	150 kGy	180 kGy	190 kGy	284 kGy
*ε*_b_ (%)	-	426.1	332.4	180.9	281.2	54.4	107.5
*σ*_b_ (MPa)	-	2.466	4.441	7.681	6.637	3.172	5.041
*σ*_m_ (MPa)	2.143	3.234	4.734	7.674	6.682	3.476	5.551
*M* 40 (MPa)	2.132	2.598	3.059	4.311	3.672	3.213	3.931

**Table 2 materials-11-02405-t002:** Results of the regression analysis of the functional dependence of *ε_b_* and *σ_b_* of the irradiated material on the magnitude of the radiation dose and statistical analysis of the quality of the found regression model.

Mechanical Characteristics	*θ_i_*	Δ*y_i_*	*m_i_*	*δ*	Δ*y*(0) (%)	SSE	R^2^	Adj-R^2^	RMSE
*ε*_b_ (%)	110	22.83	6.003	50	0.5904	8.147 × 10^−5^	1	1	9.026 × 10^−3^
	150	355.71	17.051						
*σ*_b_ (MPa)	110	2.073	3.825	2.2	2.319	4.877 × 10^−6^	1	1	1.562 × 10^−3^
	150	8.147 × 10^−5^	1.053						

**Table 3 materials-11-02405-t003:** Results of the regression analysis of the functional dependence of *σ_m_* and *M* 40 of irradiated material on the magnitude of the radiation dose and statistical analysis of the quality of the found regression model.

Mechanical Characteristics	*θ_i_*	Δ*y_i_*	*m_i_*	*δ*	Δ*y*(0) (%)	SSE	R^2^	Adj-R^2^	RMSE
σ_m_ (MPa)	77	0.617	0.287	0.015	2.143	7.104 × 10^−5^	1	1	8.429 × 10^−3^
	138	0.045	15.261						
	150	1.473	6.897						
M 40 (MPa)	77	0.272	0.321	1.591	2.708	3.737 × 10^−4^	0.9999	0.9998	3.67 × 10^−3^
	138	0.164	9.662						
	150	0.091	16.761						

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
