# Peer review of "The Effect of High-Energy Ionizing Radiation on the Mechanical Properties of a Melamine Resin, Phenol-Formaldehyde Resin, and Nitrile Rubber Blend"

_materials, 2018, doi:10.3390/ma11122405_

Round 1
Reviewer 1 Report
The work presents the effects of ionizing irradiation energy using e-beam in comparison to heating on the physical and chemical properties of the polymers where 150 kGy was found to be an optimum energy of irradiation without causing any damage to the material. Stress-strain curves with respect to dosage of the energy were presented to support the claim. The analysis was performed on melamine resin, phenol-formaldehyde resin and nitrile rubber representing a specific three-component mixture of two reactoplastics and one elastomer as representative polymer materials with different dose values 30, 100 and 300 kGy. The article is well-written and authors made a nice attempt to cover the major aspects of the study, however I believe that some areas of the manuscript need some major improvements without which the study is a little unclear. The query/comments are as below:
EB irradiation is actually a slow process in comparison to other methods, authors need to mention the time required to adequately optimize the polymer blends. As the total energy of exposure might also was a fixed exposure energy of the beam in terms of time.
Authors did not provide an explanation for a critical aspect of the results shown in Fig. 1(a), where the stress-strain curve was exponentially increasing with increase in irradiation energy from 0 to 77 and then a jump to 150 kGy happened. However, 138 kGy demonstrated higher strain and lower stress than 150 kGy and was out of the trend. Why might be the reason for that? This query makes sense if we see the curves from left to right. The color choice is so confusing, it is better to add symbols to the graph.
Even if opposite trend is the case, as authors explained that strain was higher for 77 and 138 than the stress due to partial cross-linking of the polymers, then there is a discrepancy in terms of 180 kGy which is in between 150 and 138. The authors mentioned “At 180 kGy the value σm drops to 6.682 MPa and M 40 drops to 3.67” but did not explain the reason. Again, it is strongly advised to add some different symbols in the graph to clearly identify the lines for each values and the current colors look very similar and not easy to distinguish.
What could be the possible reason of increasing and decreasing strain in Fig. 2(b) in contradiction to what authors said previously about the cross-linking. If partial cross-linking problem was till 150 kGy, then why the strain in 180 kGy increased and again decreased for 190 kGy?
Also, please correct the graph in Fig. 2(b), the strain is shown on the left Y-axis and stress on the right Y-axis. However in the graph the red bars (for stress) are shown on the left side and blue bars (for strain) are shown on the right side. Same is advised for Fig. 3(b).
Author Response
We would like to thank the reviewer for thoughtful comments and efforts.
Attached is our reply.
Thank you

Reviewer 2 Report
The effect of high-energy ionizing radiation on the mechanical properties of a melamine resin, phenol-formaldehyde resin and nitrile rubber blend
This is an excellent research paper about how ionizing radiation affect mechanical properties of polymer resins. My comments here are concerned with small errors.
1. The authors should list the information of Modulus, elongation at break, et al, in a table.
2. The explanation of Figure 1 is not enough, why 180 and 190 kGy showed so obvious difference in elongation at break?
3. Since the authors mentioned about the Tg and crystallization, DSC is needed to help better explain the stress-strain behavior. DSC can show the Tg variation and crystallization degree, et al.
Author Response

(The authors gave the same response as above.)

Reviewer 3 Report
The paperf titlede "The effect of high-energy ionizing radiation on the mechanical properties of a melamine resin, phenol-formaldehyde resin and nitrile rubber blend" by Kopal et al investigates the change occurring on blends after receiving different doses of ionizing radiation, from 77 kGy to 284 kGy.
The paper can be interesting for the readers, provided that major revisions are performed by the authors.
The major concern regards english language: extensive editing should be performed since in some parts of the paper it is difficult to understand what authors are trying to explain due to poor english.
Just to make some examples,
Line 38: "..achieve suche their properties, which fullyt meet...." should be changed in "..achieve properties meeting all the requirements..."
Line 39: "At present, one of the ways to fully controllable modification..." should be changed in "At present, a way to tailor the properties of polymeric materials..." or something similar
Line 158-159: "...that might have been caused particularly the conveyor rolls...". I cannot understand the meaning of the sentence
Line 262: "..mechanical damage, or its strenght": it is not clear what do authors mean with this sentence, since english language is confusing
in the paper "ambient temperature" should be changed with "room temperature"
Several other mistakes are scattered through the paper
Regading the presentation of results:
Materials and Methods:
1) In line 129-131 authors describe the standard uses for the blend they investigate: this part should be at the end of the introduction, not in a Materials and Methods section2) Authors do not explain where the blend comes from nor they give any useful information about it. Did they buy it? Did they get it from a company?
3) Which is the ratio between melamine/phenol-formaldheyde and nitrile rubber? This will greately affect the final properties after irradiation, since the three polymers have different behaviour
4) Besides, authors do not reveal how they prepared PS PMX3, they simply state that "The investigated PS PMX3 ewas prepared in the form of granules..." without explaining how.
5) In line 135, a pressur is given in kN: I believe that this is incorrect
6) When authors, in line 140, use ISO 37, they should anyway give an indication of the dimensions of the specimens, to help readers understanding how the experiments were performed without buying the ISO 37 dossier
When authors present the Results and Discussion:
7) In lines 192-197 authors talk about tensile modulus, ultimate strenght, tear stress etc but they only present a stress-strain graph with no indication of the values of all the mechanical parameters cited. Since they say that there is a short linear region, numbers of such parameters could help in giving an idea on how the materials are behaving in that region
8) Authors show the magnification of the 0-1% strain region. Since they state that the linear region is very narrow, have they tested also other tensile test movement speed, to check if the region can be widened?
9) Since the discussion regarding the behaviour of the specimens involves they rubbery-glassy state, authors should check glass transition temperature of the specimens, either via DSC or via DMTA.
9) Since most of the discussion is based on the fact that cross-linking and other chemical reactions arise, I suggest to perform some analyses on the samples before and after irradiation, to check which bonds are creating/changing, discussing what they see. FT-IR would be very important, but also other analyses could help
Author Response

(The authors gave the same response as above.)

Reviewer 4 Report
The effect of high-energy ionizing radiation on the mechanical properties of a melamine resin, phenol-formaldehyde resin and nitrile rubber blendThe paper presents some experimental results on the EB treatment of a terpolymer.
Indeed, the work is focused on mechanical properties only. There is no concern about other fundamental structural analyses which are necessary in order to corroborate the presented results.
Actually, all the conclusions, although they appear reasonable, are based on many suppositions.
The modelling, as claimed by authors, is a statistical analysis and no attempt to correlate fitting parameters is presented.
The preparation of samples must be described. Authors refer to a paper “Eddoumy, F.; Kasem, H.; Dhieb, H.; Buijnsters, J. G.; Dufrenoy, P.; Celis, J. P.; Desplanques, Y. Role of constituents of friction materials on their sliding behavior between room temperature and 400° C. Mater. Des. 2015 65, 179-186.” Indeed it is not clear if the material is a commercial product or it was prepared in the laboratory. Information’s on composition must be provided. Also the experimental procedure is not clear. In which way granules were obtained? Why do they use a “vulcanization hydraulic press” in order to “plasticize” the material? Which is the effect of the rolling procedure?.
Other minor issues:
Line 65 both these
Line 67 what is the meaning of “irradiation environment”
Line 482 polyethylene
The paper can be considered for publication after major revisions
Author Response

(The authors gave the same response as above.)

Round 2
Reviewer 1 Report
The effect of high-energy ionizing radiation on the mechanical properties of a melamine resin, phenol-formaldehyde resin and nitrile rubber blend
The authors have modified the manuscript as per the reviewer’s
suggestion. They have added the time required for adequate irradiation,
moreover the effects of variation in irradiation energy on resins
stress/strain curves has been further clarified. Also, there were some
minor errors in their graphs, which have been corrected and about the
discrepancy, they mentioned that would investigated in future work. The
modifications look convincing. Authors explain the requested questions
and added the respective texts in the revised version. This manuscript
can be considered for publication without further revision.
Reviewer 3 Report
Dear Authors,
the paper has been improved, and now I can recommend publication. Only two minor issues must be corrected
Line 135: I believe that PM should be PS
In Table 1, significant digits should be more uniform. In M 40 at 0 kGy there are only 2 significant digits, whereas in M 40 at 150 kGy there are 4 of them. Also for other parameters they are not uniform
Author Response
We would like to thank the reviewer for his/her insightful comments on the manuscript. Attached is the paper with the highlighted corrections.

Reviewer 4 Report
The paper can be accepted.
Author Response
We would like to thank the reviewer for his/her insightful comments on the manuscript. We corrected 2 errors in the paper.
Thank you
